# High-Efficiency of PVD Coating Process by Applying an Additional Rotation

Ivan Mrkvica *, Tomas Szotkowski, Aneta Slaninkova and Tibor Jurga

Faculty of Mechanical Engineering, VŠB–Technical University of Ostrava, 17. listopadu 2172/15, 708 00 Ostrava-Poruba, Czech Republic; tomas.szotkowski@seznam.cz (T.S.); aneta.slaninkova@vsb.cz (A.S.); tibor.jurga.st@vsb.cz (T.J.)
* Correspondence: ivan.mrkvica@vsb.cz; Tel.: +420-597-324-451

**Abstract:** This article analyzes PVD coatings (physical vapor deposition—the coating material is vaporized and deposited by sputtering or arc evaporation, and PVD coatings are applied at lower substrate temperatures and thus can be applied to a wider range of substrates) applied to samples which are located in a fixture. This fixture enables additional rotation of the sample via the coating process. The fixture allows an increase of coated tools in one batch, and therefore an increase of the current capacity of the coating machine. The introductory section of the article describes the process of product design, including its modifications. The experimental section is focused on the functionality checking of the proposed design. The coating process was carried out on a machine named INNOVA. To guarantee the correct coating application during the process, it is necessary to research the coating thickness and the chemical composition of the samples and compare these results with the results of samples which were coated without using a designed fixture. Round bars with a diameter of 10 mm were chosen as test samples. On these samples, a FUTURA monolayer was applied on a TiAlN base. Chemical composition and coating thickness were evaluated using a scanning electron microscope (SEM). Using a fixture with a fourth rotation, the same chemical composition and coating thickness were achieved as those samples which were coated in a process without the use of a fourth rotation. Therefore, it was possible to confirm a capacity increase of the coating machine.

**Keywords:** PVD (physical vapor deposition); fixture with fourth rotation; coating thickness; chemical composition of the layer

## 1. Introduction

Physical vapor deposition (PVD) refers to a variety of vacuum deposition methods. Physical processes such as sputtering and evaporation are used in PVD to generate a vapor, in the form of atoms, molecules, or ions, of the coating material supplied from a target. They are then transported to and deposited on the substrate surface, resulting in coating formation. In PVD processes, the substrate temperature is substantially lower than the melting temperature of the target material, making it feasible to coat temperature-sensitive materials. Coated tools diminish the friction and the interaction between the tool and chip and improve the wear resistance in a wide cutting temperature range [1]. Rotary holdings of coating machines are designed with a goal to coat the maximum number of tools in one cycle. Each of the coated tools (e.g., drills, milling cutters, reamers, taps) must perform an elementary motion, which is rotation around its own axis [2]. This rotation is usually realized by putting the tool into the mandrel. During every rotation, the mandrel with a tool holder is turned by a kicker. Further movements are provided by the rotation of the holder and by the rotation of the machine structure itself. The rotation of the structure is performed by the driving shaft and subsequently transmitted by a planetary gear to the mandrels (Figure 1). These movements ensure an even transfer of the coating around the

entire circumference of the coated part [3]. Any change is important from the point of view of the cutting process because it influences adhesion between the substrate and the applied coatings that accurately copy the surface structure of the cutting tool [4,5]. The so-called "shadow effect" occurs in the absence of any of these movements, which is the cause of uneven growth of the coating layers [6–8].

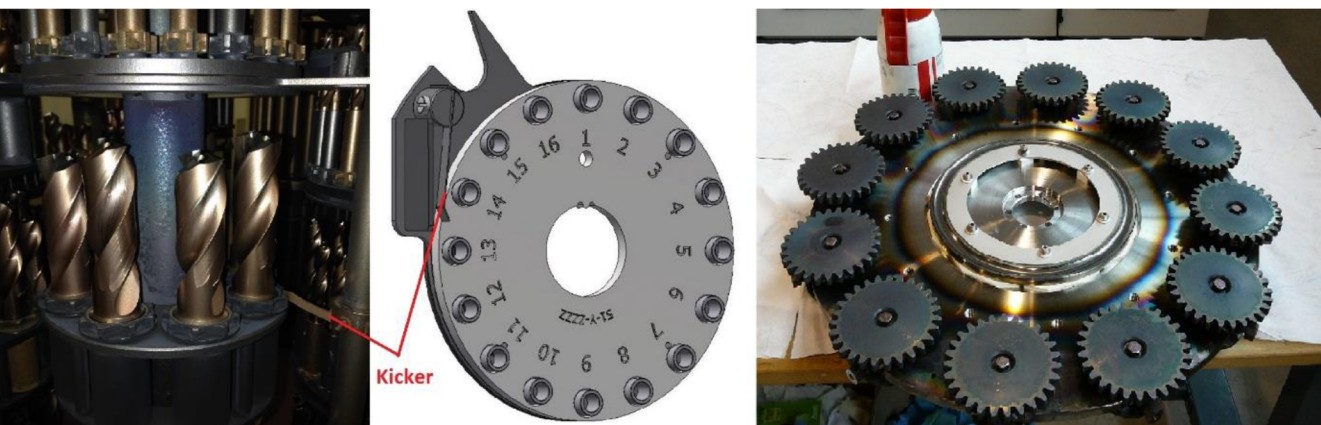

**Figure 1.** Realization of rotational movements during coating [9,10].

Uneven coating growth occurs in a shadowed location on the substrate surface. Here, the surface of the substrate is covered from the deposited target particles by another substrate or by the chamber structure itself. At this point, only a part of the deposited material is adhered, which forms an incomplete coating, and only the heat-affected zone is created on the substrate [9].

The main goal for the design of the fixture was to increase the capacity of the coating machine for some types of coated tools, e.g., mainly the economic aspect [11]. The fixture has been designed so that it can be fitted into existing stainless-steel housings without the necessity to change the rotary holder itself. These stainless-steel housings are designed with a predetermined number of positions, which could be changed according to the size of the coated substrate. It is possible to increase the chamber capacity by up to 100% for one substrate diameter [9].

The whole housing of the rotary holder is moved by planetary gear, and it is necessary to observe the maximum size of the fixture so that it does not exceed the size of the housing in which it is inserted [12]. The maximum size of the fixture depends on the diameter of the housing, and it is therefore possible to use the innovation only for a smaller spectrum of the coated tool size. The diameter of the stainless-steel housing with positions for the substrates is the same for all substrate sizes. For these housings, only the number of positions changes [10]. The number of positions cannot be increased without adding motion using auxiliary rotation. It is necessary to synchronize the movement of the substrate in the fixture (the fourth rotation) with the planetary movement of the rotary holder and therefore with all three basic rotations. This fourth rotation ensures smooth movement and even coating production on the substrate, which is an essential condition for the PVD method [13,14].

## 2. Progressive Modifications in the Design of the Fixture

The fixture, which allows rotation about the fourth axis, is a geared transmission with four rotating hinges for the location of the coated substrate. The construction was designed from chrome-nickel austenitic steel EN 1.4301 (X5CrNi18-10) for its resistance to corrosion and high temperatures (Tables 1 and 2). An important criterion in steel selection was its good electrical conductivity for implementation of the coating process.

**Table 1.** Mechanical properties of steel EN 1.4301 [15].

| Mechanical Properties | Values |
|---|---|
| Ultimate tensile strength, Rm | 520–720 N/mm$^2$ |
| Yield strength, Rp 0.2 min | 210 N/mm$^2$ |
| Ductility, A 80 mm | 45% |

**Table 2.** Chemical elements of steel EN 1.4301 [15].

| Chemical Element | Values (%) |
|---|---|
| C | <0.07 |
| Si | <1.00 |
| Mn | <2.00 |
| P | max. 0.045 |
| S | max. 0.030 |
| N | <0.11 |
| Cr | 17.00–19.50 |
| Ni | 8.00–10.50 |

The first step in the implementation of the fixture was to remove the bottom of the geared part used in the current machine structure to rotate the entire machine structure (Figure 2). This change was initiated by the possibility of using a rotating hinge in stainless-steel housing, where the fixture can be placed. Part of this rotating hinge is gearing, and therefore its occurrence on the fixture is not necessary. However, the samples (substrates) were not coated around the perimeter and had a shadow effect fault because there was insufficient rotation of the samples on the fixture.

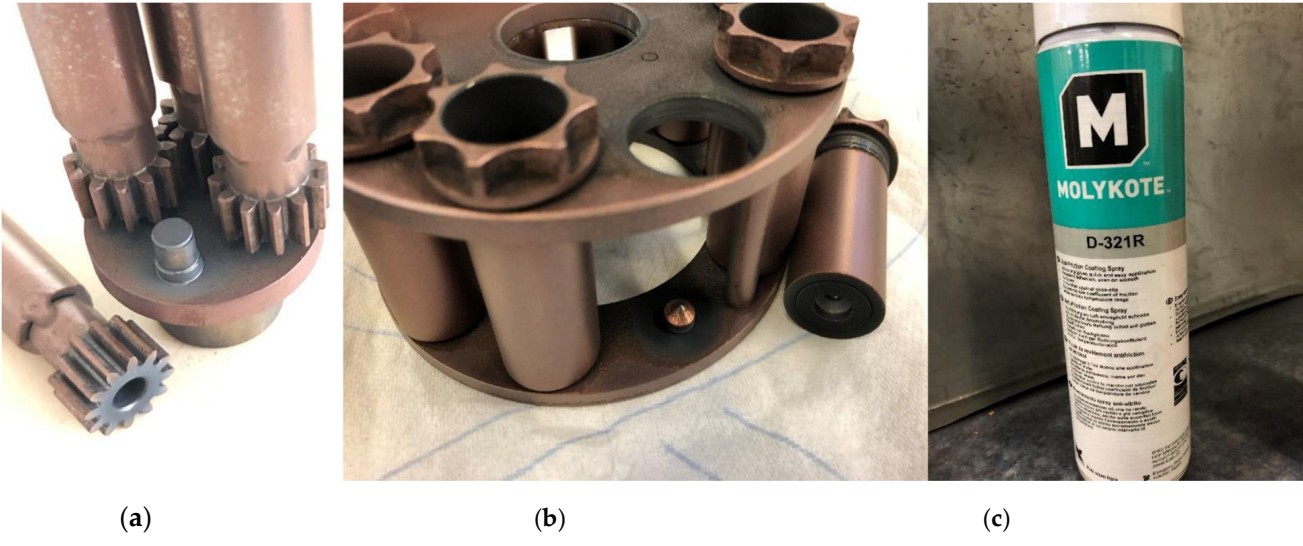

(**a**)　　　　　　　　(**b**)　　　　　　　　(**c**)

**Figure 2.** (**a**) Mounting of the fixture axis, (**b**) bronze pin (a steel washer was inserted here to increase friction), and (**c**) lubricant used for friction reduction.

For economic and design reasons, the gear wheels were not fitted with bearings. There was little clearance between the gears and the axes of the fixture, which was sufficient for the movement of these gears. However, after loading by substrates, the wheels rubbed over the entire face area on the base of the fixture. This friction made it impossible to rotate the rotary hinges around its own axis, as it was greater than the friction between the mounting of the fixture itself and the stainless-steel housing. A certain solution was to create a shoulder on the axis, which had a larger diameter than the diameter of the hole in the gear (Figure 2a). Furthermore, it was necessary to achieve greater friction between the fixture holder and the stainless-steel housing into which the fixture is inserted and

less friction between the fixture base and the gearing. These conditions were achieved by inserting a steel washer between the fixture holder and the stainless-steel housing (this part of the structure is made of bronze) (Figure 2b).

A layer of Molykote D-321 R lubricant (Figure 2c) was applied between the two components to achieve optimal friction between the gear and the fixture axis. This lubricant does not lose lubrication capabilities up to 450 °C and is ideal for reducing friction in a vacuum [16]. This lubricant replaced the function of the slide bearings between the gears and the fixture, thus reducing the production cost of the fixture itself. This lubricant is always added whenever the machine is set up.

The last adjustment that was made to the fixture was the adjustment of the gearing. With the third version of the fixture, it was already possible to coat the samples around the entire circumference, and the fourth rotation was achieved. This added rotation removed the unwanted shadow effect. During monitoring of the process, another undesirable movement has been found that can make it difficult to coat around the entire circumference of the substrate. This is a situation where a kicker which rotates with a gearing hits the first gear and spins the entire fixture. If the kicker hits gear number 2, the entire gearing set rotates in the opposite direction. This phenomenon can cause layers of deposited material to become attached to the same sides of the substrate surface, resulting in an uneven coating around the circumference. A graphical representation of this phase of the fixture design is shown in Figure 3.

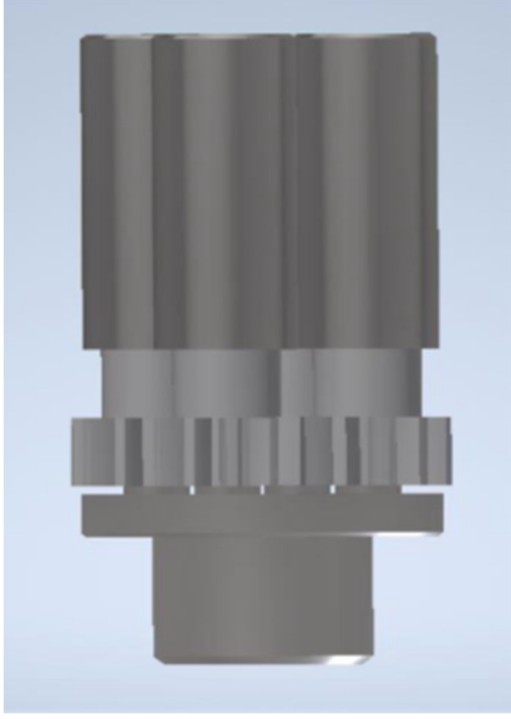

**Figure 3.** Fixture version without gearing modification.

The negative aspect was suppressed by the final design modification. For gears 2 and 4, the tooth length was halved to 4 mm (Figure 4). This modification prevents the kicker from hitting two gears next to each other. Thus, the samples rotate in only one direction, and this ensures that the coating grows evenly around the entire circumference of the inserted substrates.

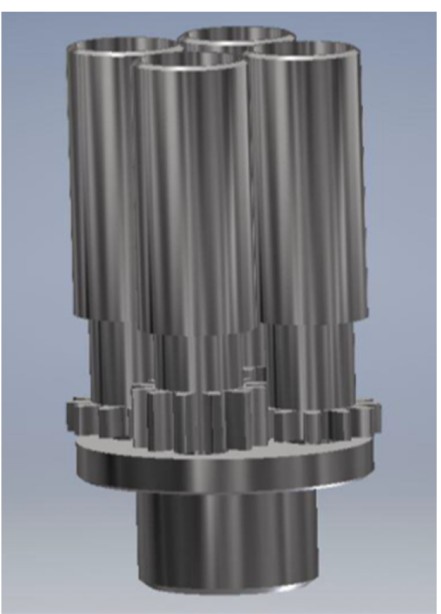

**Figure 4.** Fixture version with gearing modification.

## 3. Measurement and Evaluation of the Coated Samples: Coat Thickness and Chemical Composition

Cylindrical rods were used as test substrates. The material was EN 1.4307 (X2CrNi18-9) [17], often used in industrial practice [18,19]. The samples were cut from a blank with a diameter of 12 mm to a final diameter of 10 mm, with a surface roughness Ra = 0.8 μm. The samples simulate rotary tools, e.g., drills, reamers, or taps, with the same diameter. The modified samples were placed into a fixture for quadruple rotation and also in a free position in stainless-steel housing, allowing only triple rotation of the substrate. For the most appropriate comparison, all samples were placed into the same stainless-steel housing (Figure 5). During experiments, three sets (3 × 6 samples) of samples were used, and values are the average of the measured values.

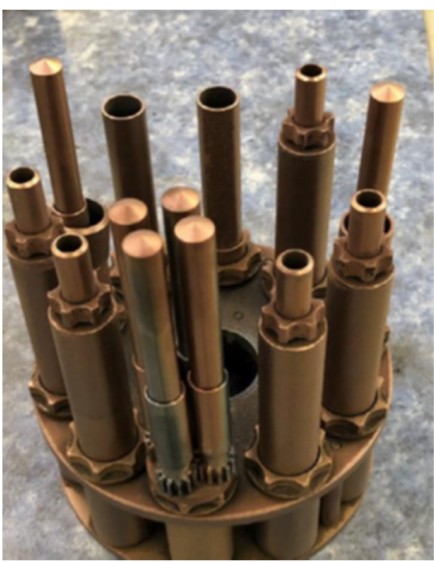

**Figure 5.** Method of samples' location in stainless-steel housing.

After the coating process, the coated substrates were cut into 3 mm-thick samples with EDM (to eliminate the heat-affected area at the cutting point). The cut samples were pressed into a synthetic resin (ATM Duplast, DUPLAST SPA, GIOIA DEL COLLE, BARI,

ITALY–see Figure 6) and a metallographic cut was made. Samples were evaluated using a QUANTA 450 FEG scanning electron microscope (SEM, FEI Company, Brno, Czech Republik) [20]. Samples were also evaluated on other optical microscopes, namely the Keyence VHX 6000 [21] and the Alicona Infinite Focus [22]. However, it was not possible to determine all the required properties of the substrate and the deposited material with these devices. In addition, it was not possible to determine with certainty the correct coating thickness in the images from both optical microscopes. The boundary between the coating and the substrate could not be precisely defined in these images, and therefore the thickness measurement by these instruments was imperfect (Figure 7). Using the selected electron microscope, it was possible to accurately determine the thickness of the coating and its chemical composition.

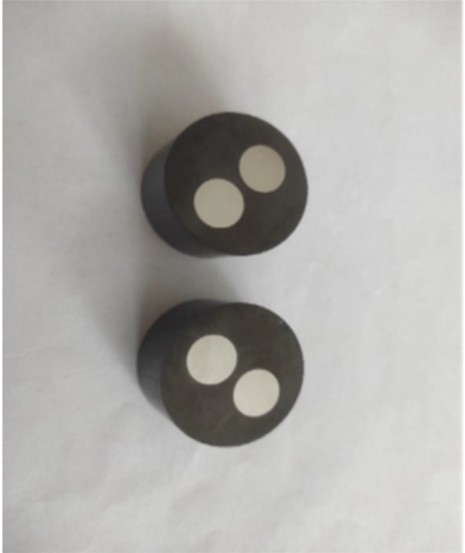

**Figure 6.** Metallographic cut of samples.

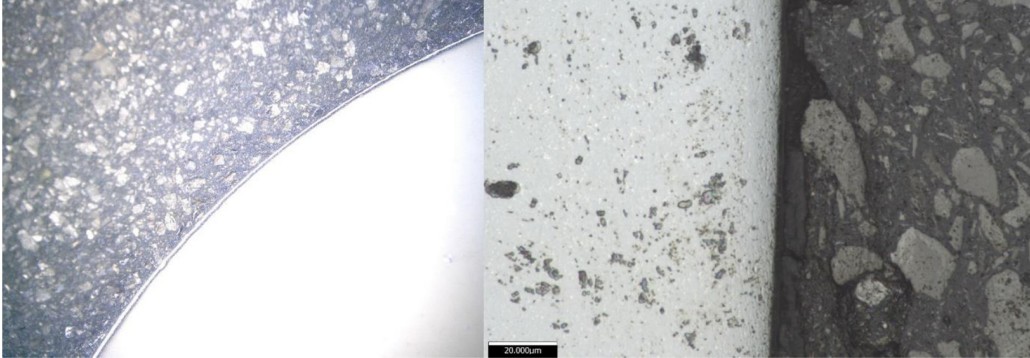

**Figure 7.** Images of samples from optical microscopes (from left, Keyence VHX 6000, Alicona Infinite Focus).

The coating was applied to the substrate to a tolerance of 1–3 μm. The measured sample was divided into 4 parts (quadrants). In these parts of the sample, the thickness of the deposited material was measured, and the chemical composition of the substrate and the coating was determined. This chemical composition of the coating was determined for each sample, because if the layer thickness met the prescribed tolerance and the coating would be poorly applied due to the difference in chemical composition, it would also be a defective piece [23]. Destruction of the coating layer was also observed in the samples, which could be caused by grinding of the sample during the formation of a metallographic cut. Abrasive wear in some places was characterized by the disintegration of the coating

layers. Parts of these layers were further pressed into the synthetic resin during the metallographic cut (Figure 8). This deformation did not affect the measurement.

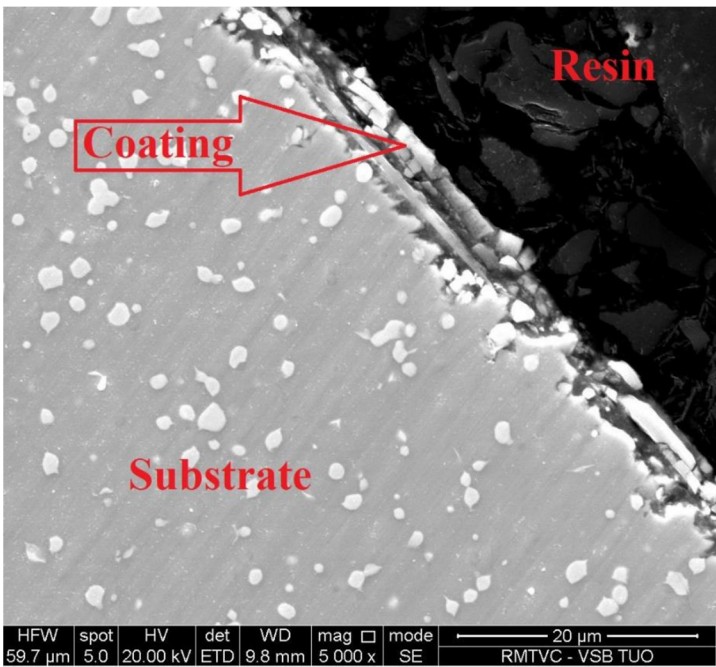

**Figure 8.** Deformation of coated layers due to abrasive wear during metallographic cutting.

The chemical composition of the coating (Figure 9) was determined using the so-called peak area integration, which is a method of chromatographic data evaluation. In Figure 9, the horizontal axis is the value of the X-rays used (in keV), and the vertical axis is the intensity. The chemical composition was then determined using this method.

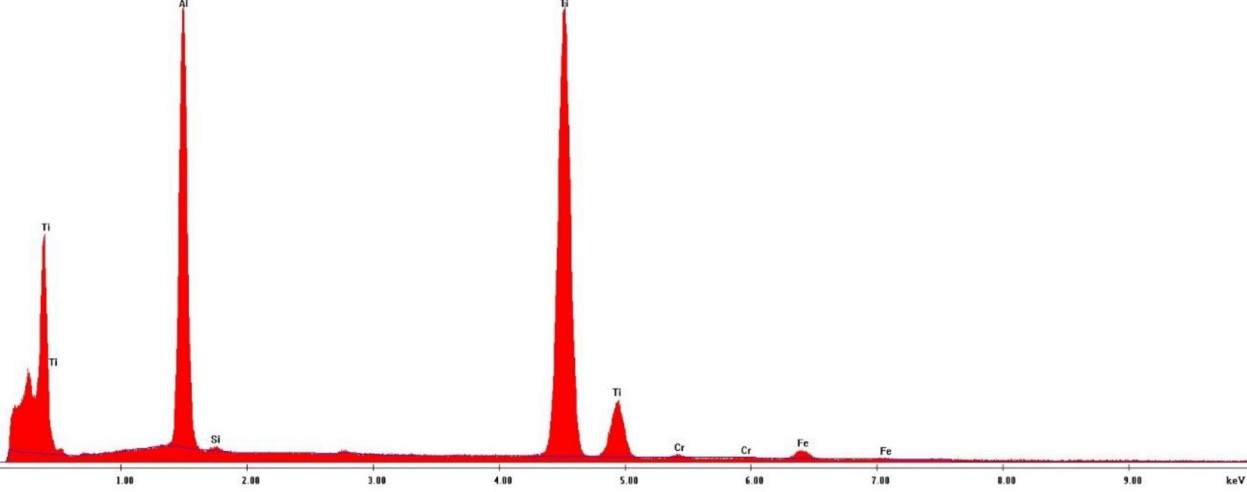

**Figure 9.** Peak surface integration for chemical composition of the coating.

## 4. Discussion of Results

The first tested sample was a substrate which was coated separately in a rotary holder with one position, but in the same stainless-steel housing as the fixture, and therefore the same coating parameters for all substrates were guaranteed.

As can be seen from Figure 10, in all measured quadrants, the coated thickness was within a specified tolerance of $2 \pm 1$ μm. Chemical analysis is important from the point of view that the content of chemical elements in this coating was used as a standard for

research and a comparison of further coated substrates. Using chromatographic evaluation of the data, the weight and atomic percentage of chemical elements in the coating was determined (Table 3).

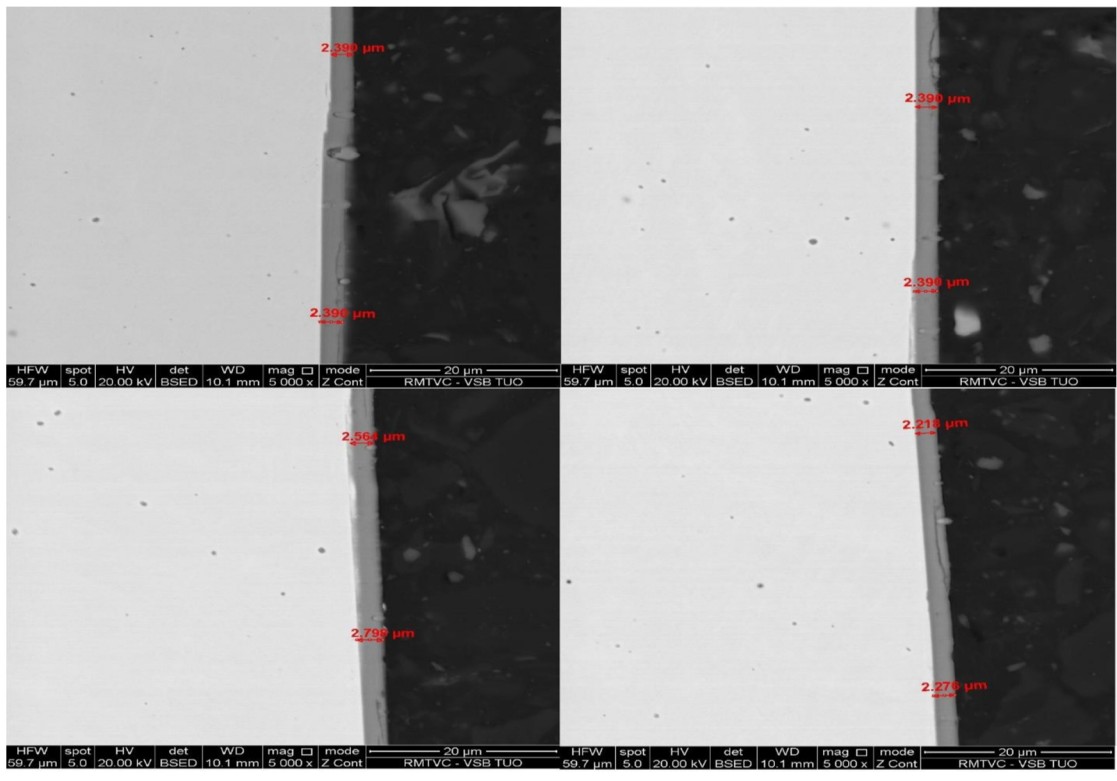

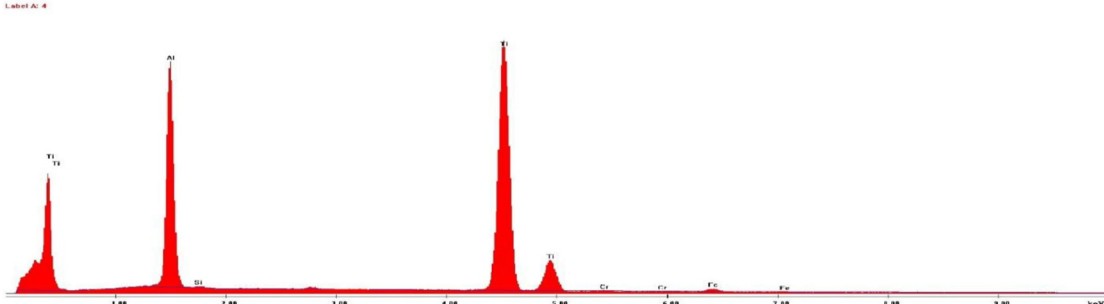

**Figure 10.** Measurement of the coating thickness of a substrate coated separately, without the addition of a fourth rotation.

**Table 3.** Chemical composition of the coating applied in the classical way.

| Chemical Element | Weight Percent (%) | Atomic Percent (%) |
| --- | --- | --- |
| Al | 29.3 | 42.5 |
| Ti | 69.1 | 56.4 |
| Cr | 0.4 | 0.3 |
| Fe | 1.2 | 0.8 |

Test substrates were also coated by using the first version of the fixture. During this test, there was a shadow effect. The samples were not coated around the entire circumference within the tolerance (Figure 11). In Figure 11, it is possible to see the measured places where the layer thickness exceeded 3 μm. The coat was deformed and cracked in these

places. In Figure 12, the thickness was below 1 μm. This part of the substrate has been shielded and the applied coating did not reach the required values. The values for the chemical composition of the sample were not significantly different than the well-coated piece (Table 4). It can be concluded that the coat was chemically unchanged during the shadow effect, but the coating elements were deposited unevenly on the entire surface.

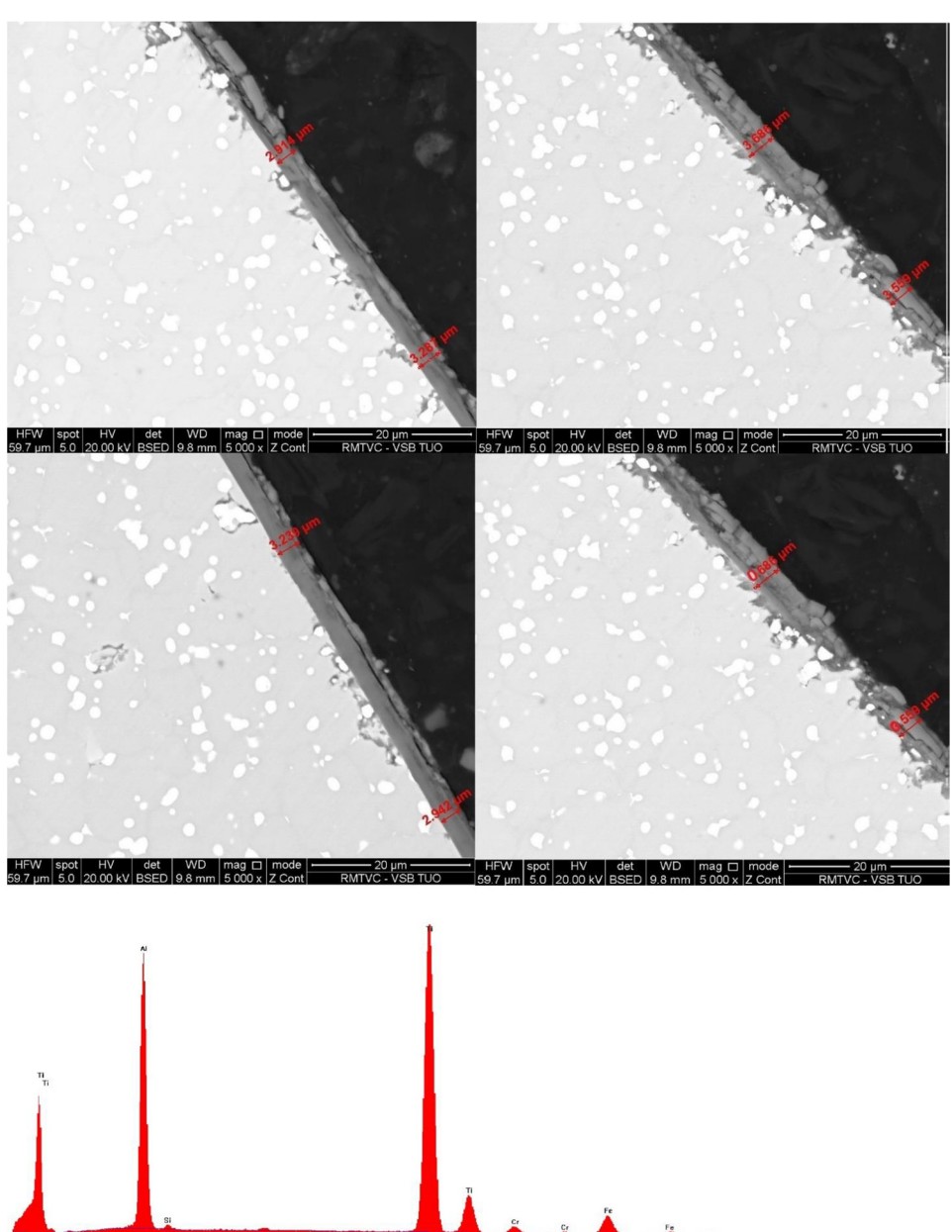

**Figure 11.** Measurement of the coating thickness of the substrate coated with the fourth rotation fixture.

**Table 4.** Chemical composition of the coating applied to the substrate with an added fourth rotation.

| Chemical Element | Weight Percent (%) | Atomic Percent (%) |
|:---:|:---:|:---:|
| Al | 28.6 | 42.5 |
| Ti | 64.3 | 54.2 |
| Cr | 1.6 | 0.8 |
| Fe | 5.5 | 2.5 |

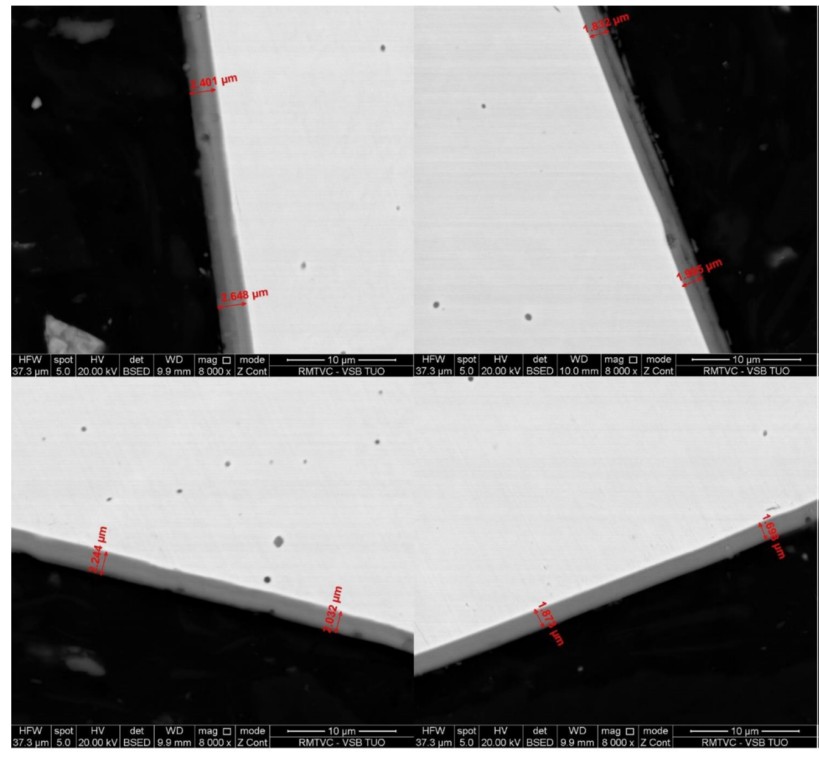

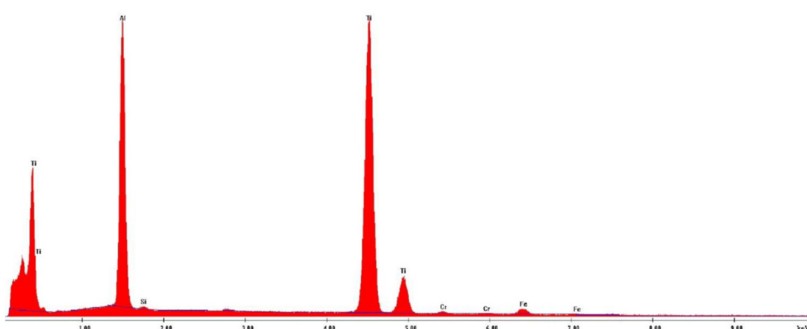

**Figure 12.** Measurement of the coating thickness of the substrate coated with the fourth rotation fixture after finishing.

After the development of the fixture construction, in the fourth and final version, the substrate was coated evenly over the entire surface (Figure 12). The thickness of the deposited material on the surface of the substrate was on average 2 μm according to all measured samples. The smallest measured thickness was 1.698 μm and the largest was 2.648 μm. This measurement confirmed that with the latest version of the fixture, it is possible to coat the substrate using the fourth rotation with the same quality as in the three-rotation process. This quality was confirmed not only by the measured coating thickness, but also by the chemical composition, which was comparable to the first sample (Table 5).

**Table 5.** Chemical composition of the coating applied to the substrate with an added fourth rotation after finishing the fixture construction.

| Chemical Element | Weight Percent (%) | Atomic Percent (%) |
|:---:|:---:|:---:|
| Al | 31.8 | 45.4 |
| Ti | 64.9 | 52.3 |
| Cr | 0.7 | 0.5 |
| Fe | 2.6 | 1.8 |

## 5. Conclusions

PVD coating technology is a very popular surface treatment method. As a result of an increasingly progressive market, it is necessary to constantly improve the methods of production of these coatings and increase the amount of production to remain competitive. The proposed fixture underwent an evolution, and samples were coated with each version. In the initial versions, there was a shadow effect which caused an uneven growth of the deposited layer. Differences in the thickness of the coating in the measured quadrants were caused due to imperfect movement of the fourth rotation. These shielded surfaces bound only part of the deposited material, only about 0.6 μm-thick on average. The sides of the substrate, which were turned towards the targets and the movement of the deposited material, trapped a larger amount of particles, and the coated layer reached an average thickness of 3.6 μm. Both of these values are out of the specified tolerance. These samples showed poor adhesion between the coated layers. These layers subsequently detached in some places, creating depressions and craters. The increasing thickness of the coating led to a decrease in the strength.

After modifications of the fixture design, the fourth rotation was added to the process. After these changes, the substrates had a quality comparable to the substrate coated separately with only 3-fold rotation. The design of the fixture with reduced friction ensures added rotation and guarantees that there are no places with a shadow effect. The deposited material was applied continuously and the grain growth of the individual elements of the coating was constant. The coating had an average thickness of 2 μm around the entire circumference, with a maximum deviation of 1 μm. The coating thickness of the last sample was therefore within the manufacturing tolerance. This result proves the possibility of using 4-fold rotation for this coating method.

The results of this research can be implemented in production from the point of view of optimization of the coating process. The capacity of the coating chamber is variable due to the possibility of coating different diameters of the substrate during one process. However, if the capacity of the chamber relates to the substrate examined above (with diameter of 10 mm), it is possible to determine it precisely. Stainless-steel housings for the above-mentioned diameter contain 20 positions for the substrate. Five pieces of this housing can be placed on one mandrel, and the whole structure contains ten mandrels. This means that, for a diameter of 10 mm, there are 1000 positions in a specific chamber with 3-fold rotation. If it includes a process with 4-fold rotation under the same conditions, then when coating substrates with a diameter of 10 mm, it is possible to achieve a capacity increase of 100%. Stainless-steel housing with a substrate diameter of 18 mm could be used. This stainless-steel case contains 10 positions, and it is also possible to place 5 of these on one mandrel of the structure and 10 mandrels on the whole structure. However, for this case, a fixture with added rotation with 4 extra positions was designed, which means 40 positions for the substrate per stainless-steel housing and a total of 2000 positions per charge into the coating chamber (Figure 13).

Summary:

1. The results proved the possibility of using a fourth rotation for this coating method.
2. It has been proven that by adding a fourth rotation to the PVD process, it is possible to coat tools to the same quality as tools in a three-rotation process.
3. The fixture with the fourth rotation positively influenced production by increasing the production batch capacity for substrate dimensions. This variant is suitable for large-scale production of coatings for cylindrical tools.

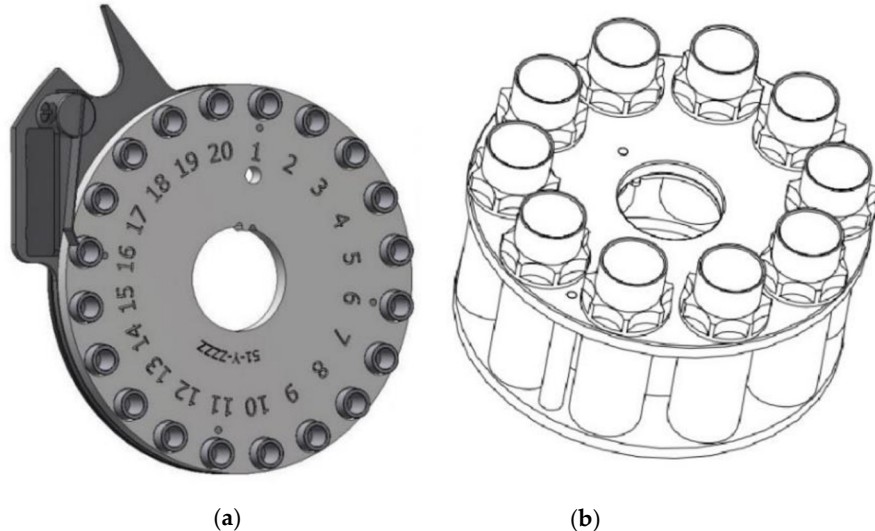

(**a**)  (**b**)

**Figure 13.** Comparison of two stainless-steel housings: (**a**) housing for 20 positions and (**b**) housing for 10 positions—use with a coating fixture [10].

**Author Contributions:** Conceptualization, I.M. and T.S.; methodology, I.M. and T.S.; validation, I.M., T.S., A.S. and T.J.; investigation, A.S.; resources, I.M. and T.S.; data curation, T.S., A.S. and T.J.; writing—original draft preparation, I.M. and T.S.; writing—review and editing, A.S. and T.J.; supervision, I.M. All authors have read and agreed to the published version of the manuscript.

**Funding:** This work has been performed in connection with projects: Specific Research Project SP 2022/50 and SP 2022/131, financed by the Ministry of Education, Youth and Sports of the Czech Republic.

**Institutional Review Board Statement:** Not applicable.

**Informed Consent Statement:** Not applicable.

**Data Availability Statement:** Data available in a publicly accessible repository.

**Conflicts of Interest:** The authors declare no conflict of interest. The funders had no role in the design of the study; in the collection, analyses, or interpretation of data; in the writing of the manuscript, or in the decision to publish the results.

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
