# Peer review of "High-Efficiency of PVD Coating Process by Applying an Additional Rotation"

_coatings, doi:10.3390/coatings12060834_

Round 1

Reviewer 1 Report

The manuscript may be recommended for publication if the authors address the following comments in the revision:

- the title should present what kind of contribution is described in the manuscript and which factor enhances productivity

- the abstract and conclusion should also present what and how influences the process

- PVD abbreviation should be explained in the abstract and in the text of the manuscript

- The experimental part should be presented separately for better readability (including materials, procedures such as sample preparation for measurements, techniques with applied parameters and apparatus type).

- The chemical composition of the samples was given to the second digit after the decimal place, probably the results were rewritten from the device, but in fact this technique does not have this measurement accuracy. The result presentation should be reconsidered.

-The introduction begins with descriptions of the coated tools. The type of tools should be precised.

Reviewer 2 Report

Congratulations. The work is good, however, there are some concerns about your work, which can be resolved, improving the work and your understanding. Please see the attachment.

Reviewer 3 Report

This article, almost in the same form, but under a different title ("PVD coating of tools by adding a fourth rotation") and with a different composition of authors (Tomas Szotkowski, Filip Kowalovski and Ivan Mrkvica) was submitted to Coatings in 2021 (coatings -1303213). Then I did not recommend the article for publication and gave a number of recommendations that were absolutely not taken into account by the authors. I think that the authors do not quite correctly in this case. I wrote a few comments, but stopped because my previous comments remained without any response from the authors. In general, the article remains weak, written in poor English and cannot be recommended for publication. Here are the comments I made. My earlier comments may have been archived in coatings-1303213. If necessary, I am ready to send them again.

Comments:

English at a low level! I recommend correction - pity with the help of a professional language service.

The abstract must be completely changed. The abstract is not a summary of the content of the article. Abstracte's task is to present the main ideas and results of the research in a concentrated form.

SEM is absolutely not suitable as a keyword. Keywords should describe the scientific field, not research methods.

"chemical composition of the coating" is also not suitable - too long and general, besides the word "coating" is used three times in the keywords.

"Coated tools compared with uncoated ones offered better protection against mechanical and thermal loads..." is a false statement!

Reviewer 4 Report

Dear Authors,

Congratulations on your work, which is focused on a very interesting subject. As any other paper in this phase, there are some amendments to do, whose can improve the overall quality of your paper. Thus, I'm providing below some comments and suggestions, trying to collaborate by this way in improving your paper:

  1. The Abstract doesn't clearly state the literature gap found, as well as the main motivation to develop this work. Thus, please clearly state the gap found in the literature in the Abstract, Introduction and Conclusions. The mains goals are also not clear in the Abstract.
  2. The novelty brought by your work is also not properly pointed out. Thus, please state clearly the novelty that your paper represents for the scientific community, stating as well if your contribution is exclusively scientific or if there was some practical motivation behind the development of your work. Any industrial application based on this work should also be pointed out.
  3. The title is Ambiguous, seeming a "management paper". Please give a more technical sense to the title.
  4. The Abstract sums the work done, but seems a set of individual sentences/ideas. Please give them an adequate flow/sense.
  5. Please highlight your main contribution with quantitative data. Please refer the gain in productivity in %, referring in each operations the gains are more evident (mainly in the fixture system developed).
  6. The paper should be reorganized in order to show the concept and, after that, the case study. However, in section 2, you are starting by describing the case study (validation), not the concept. Please change the order, giving a more theoretical contribution, and validating it through a case study (my students do the same mistake...).
  7. Please proofread your paper avoiding small mistakes such as in page 138: "... quad-rants...". Revise all paper, please.
  8. Please use always a free space between units and values (2±1μm, for example, line 159). Revise all paper, please.
  9. The EDS spectra of Figures 10  and 11 need to be improved. The information givemn is scarce.
  10. The discussion is poor. Please compare your results with other works previously done in the same path.
  11. Regarding the Conclusions, your main gain is in quality of the coatings, not in productivity. Thus, the title needs to be amended, in fact. Moreover, the Abstract, Main Goals, NOVELTY, main contribution brought by your work and reorganization of your paper are MANDATORY.

Thus, please take this opportunity to improve, in fact, you work, saving it of rejection. Please reorganize and complete your paper following the Reviewers' comments.

Best wishes.

Kind regards,

Round 2

Reviewer 1 Report

The text of the manuscript was improved. I recommend only a minor revision.

- The current title is too long. My suggestion: “High-efficiency of PVD coating technique by applying a 4-rotation process” or “High-efficiency of PVD coating process by applying an additional rotation”.

- The explanation of using EDS for evaluation of coat thickness was clear, but still is not clear why the authors implemented the study the results at the second decimal place for EDS results if the technique accuracy does not allow that kind of presentation – chemical composition given in the Tables of 3, 4 and 5.

- PVD abbreviation should be explained in the abstract, not for keywords. I suggest using the whole name in the abstract, with or without abbreviation. 

Author Response

Thank you for your review.

- The tittle was adjusted according to recommendation.

- EDS results were adjusted.

- PVD abbreviation was explained in the abstract.

Best Regards

prof. Ivan Mrkvica, Ph.D., MSc.

VŠB – Technical University of Ostrava

Faculty of Mechanical Engineering

Department of Machining, Assembly and Engineering Metrology

Reviewer 2 Report

The authors have made a sufficient improvement in their work. In my opinion, the article meets the minimum requirements to be published.

Congratulations.

Author Response

Thank you for your review.

Best Regards

prof. Ivan Mrkvica, Ph.D., MSc.

VŠB – Technical University of Ostrava

Faculty of Mechanical Engineering

Department of Machining, Assembly and Engineering Metrology

Reviewer 3 Report

Fig 2 - I do not see the point in the image of the lubricant packaging. If there is any important information on the packaging, they can be presented in the text.

Figure 1 is partially duplicated in Figure 13. There is no point in repeating identical images.

Figure 7. A distinguishable scale bar should be provided for each image.

All images should be labeled (a), (b) ... not "left to right"

The conclusion should be substantially shortened and focused. It is advisable to list the main results 1.... 2... 3.... The images are not very appropriate in the Conclusion (especially since they largely repeat the previously presented images).

Author Response

Thank you for your review.

  1. On the lubricant packaging are no important informations for the text of article.
  2. Part of figure 1 is repeated bay way of better illustration about economical consideration in conclusion of article.
  3. A distinguishable scale was not used in figure 7 because coat evalution by optical microscopes was not article objectives.
  4. Label of images was adjusted.
  5. The conclusion was complemented by main results.

Best Regards

prof. Ivan Mrkvica, Ph.D., MSc.

VŠB – Technical University of Ostrava

Faculty of Mechanical Engineering

Department of Machining, Assembly and Engineering Metrology

Reviewer 4 Report

Thank you for having properly addressed my comments.

Kind regards. 

Author Response

(The authors gave the same response as above.)
